# Cranial Ultrasound Abnormalities in Small for Gestational Age or Growth-Restricted Infants Born over 32 Weeks Gestation: A Systematic Review and Meta-Analysis

**DOI:** 10.3390/brainsci12121713

**Published:** 2022-12-14

**Authors:** Charlene Roufaeil, Abdul Razak, Atul Malhotra

**Affiliations:** 1Monash Newborn, Monash Children’s Hospital, Melbourne, VIC 3168, Australia; 2Department of Paediatrics, Monash University, Melbourne, VIC 3168, Australia

**Keywords:** fetal growth restriction, intrauterine growth restriction, intraventricular haemorrhage, moderate-late preterm, neonate, periventricular leukomalacia, term

## Abstract

Aim: To perform a systematic review and meta-analysis of existing literature to evaluate the incidence of cranial ultrasound abnormalities (CUAs) amongst moderate to late preterm (MLPT) and term infants, affected by fetal growth restriction (FGR) or those classified as small for gestational age (SGA). Methods: A systematic review methodology was performed, and Preferred Reporting Items for Systematic Review and Meta-Analyses (PRISMA) statement was utilised. Descriptive and observational studies reporting cranial ultrasound outcomes on FGR/SGA MLPT and term infants were included. Primary outcomes reported was incidence of CUAs in MLPT and term infants affected by FGR or SGA, with secondary outcomes including brain structure development and growth, and cerebral artery Dopplers. A random-effects model meta-analysis was performed. Risk of Bias was assessed using the Newcastle-Ottawa scale for case–control and cohort studies, and Joanna Briggs Institute Critical Appraisal Checklist for studies reporting prevalence data. GRADE was used to assess for certainty of evidence. Results: Out of a total of 2085 studies identified through the search, seventeen were deemed to be relevant and included. Nine studies assessed CUAs in MLPT FGR/SGA infants, seven studies assessed CUAs in late preterm and term FGR/SGA infants, and one study assessed CUAs in both MLPT and term FGR/SGA infants. The incidence of CUAs in MLPT, and late preterm to term FGR/SGA infants ranged from 0.4 to 33% and 0 to 70%, respectively. A meta-analysis of 7 studies involving 168,136 infants showed an increased risk of any CUA in FGR infants compared to appropriate for gestational age (AGA) infants (RR 1.96, [95% CI 1.26–3.04], *I*^2^ = 68%). The certainty of evidence was very low due to non-randomised studies, methodological limitations, and heterogeneity. Another meta-analysis looking at 4 studies with 167,060 infants showed an increased risk of intraventricular haemorrhage in FGR/SGA infants compared to AGA infants (RR 2.40, [95% CI 2.03–2.84], *I*^2^ = 0%). This was also of low certainty. Conclusions: The incidence of CUAs in MLPT and term growth-restricted infants varied widely between studies. Findings from the meta-analyses suggest the risk of CUAs and IVH may indeed be increased in these FGR/SGA infants when compared with infants not affected by FGR, however the evidence is of low to very low certainty. Further specific cohort studies are needed to fully evaluate the benefits and prognostic value of cranial ultrasonography to ascertain the need for, and timing of a cranial ultrasound screening protocol in this infant population, along with follow-up studies to ascertain the significance of CUAs identified.

## 1. Introduction

### 1.1. Background

Cranial ultrasonography is an easily accessible, well-accepted, first-line imaging modality that is widely used to screen preterm and high-risk infants for brain injury in the neonatal period [1,2]. It is best utilised for the detection of major cranial abnormalities such as haemorrhagic or ischaemic lesions [3,4]. Preterm neonates born less than 32 weeks’ gestation are most susceptible to brain injury due to differences in the vascular, cellular, and anatomical features of the premature brain, whereas infants born moderate to late preterm (MLPT) (32–36^+6^ weeks’ gestation) and term age, have not been widely studied and are likely to have a different pattern of brain injury [5,6,7]. Current guidelines suggest serial cranial ultrasonography as a screening tool in very preterm infants (<32 weeks) or very low birth weight infants (<1500 g), in addition to ‘high risk’ neonates (for example infants born with hypoxic-ischaemic encephalopathy, congenital infections or central nervous system infection) falling outside of these parameters [3,8,9]. The definition of ‘high-risk’ infants is however often subjective and depends on individual neonatal unit practices and resources available. 

While preterm infants born less than 32 weeks are at the highest risk of brain injury as evident on cranial ultrasonography, most preterm infants are in fact born between 32–36^+6^ weeks gestation. Infants born moderate preterm, between 32–33^+6^ weeks, and late preterm, between 34–36^+6^ weeks, comprise 80% of preterm births [10]. While this cohort has historically been seen as low risk, evidence is mounting that infants born moderate to late preterm (MLPT) are also at risk of long-term neurodevelopmental concerns, with an increased risk of cognitive, motor, and behavioural abnormalities in school age children and a three-fold increased risk of language delay compared to term-born controls at two years corrected age [11,12,13,14].

Additionally, birth weight has significant implications for morbidity and mortality. Infants born small, namely those affected by fetal growth restriction (FGR) or born small for gestational age (SGA) are at increased risk of neurodevelopmental complications [15]. FGR indicates the inability of a fetus to meet their expected growth potential, due to a pathological process leading to a lack of oxygen and nutrients, most commonly resulting from placental insufficiency [15]. SGA, defines infants born less than the 10th centile for weight for that gestation [16] but fails to differentiate between infants that are constitutionally small or affected by pathology causing FGR. Infants with FGR have been identified as at risk for neurodevelopmental complications, including an increased risk of cerebral palsy as compared to infants born with a weight appropriate for gestational age (AGA) [17,18,19]. They also have difficulties with attention, memory, speech, and cognition in school aged children [20,21,22]. The long-term neurological sequelae seen in infants with FGR are explained by structural and functional changes to the brain that occur due to the hypoxic state caused by placental insufficiency [18]. They also have a smaller head circumference, reduced intracranial volume, decreased cortical grey matter, and altered profile of white matter myelination [23,24,25,26].

The neurodevelopmental complications of FGR are often confounded by prematurity, thus, growth restriction amongst premature infants may be assumed to pose a greater risk of intracranial pathology due to a combination of pathogenic factors. Despite the increasing evidence of poorer neurological outcomes in FGR/SGA infants, there is sparse literature evaluating the benefit and utility of cranial ultrasound screening amongst the MLPT and term growth-restricted/SGA infants. This raises the question of the yield and benefit of screening cranial ultrasounds in theses infants to help determine pathology that may underpin their neurodevelopmental complications. To date, there have generally been inconsistent findings around the incidence of abnormalities found in this population [19,27]. This may be attributable to the heterogeneity associated with infants diagnosed with FGR. The difficulty of defining these infants is confounded when birth weight is the sole criteria for growth restriction, this fails to distinguish between infants that are affected by growth restriction and those born constitutionally small. This has been a contentious issue and historically FGR has been inconsistently defined through the literature [28].

### 1.2. Aim

This study aimed to perform a systematic review and meta-analysis of the literature for the incidence of cranial ultrasound abnormalities (CUAs) amongst MLPT and term infants, affected by fetal growth restriction or classified as SGA. Given FGR is inconsistently defined throughout the literature, Fetal growth restriction (FGR)/Intrauterine growth restriction or SGA (≤10th centile for birthweight) have been used as surrogate markers for growth restriction. Hence, for the purpose of this study, FGR refers to infants classified as either FGR/IUGR or SGA unless specified.

## 2. Methods

A systematic review methodology was chosen to investigate what is known about the incidence of abnormal cranial ultrasonography findings amongst infants born from 32 weeks to term gestation and affected by FGR and identify gaps for further research. The (Preferred Reporting Items for Systematic Review and Meta-Analyses) PRISMA statement (Code: CRD42022341804) was utilised [29]. Results were presented as a meta-analysis and synthesis of evidence. The research protocol was registered on PROSPERO (CRD42022341804).

### 2.1. Eligibility Criteria

(1)Inclusion criteria

Studies reporting any cranial ultrasonography abnormalities amongst growth-restricted and/or SGA infants published were included. No search filters for publication type or date range were used to ensure a comprehensive search. The results were limited to studies published in English/with English translation and those conducted in humans. Studies were manually reviewed to see if infants from 32 weeks’ gestation onwards were included. Case-control, cohort studies and descriptive studies were included. Studies that presented findings for neonates born out of the gestational age criteria but provided subgroup analysis for relevant study population were included. Authors were contacted if subgroup analysis was not available, and if data was provided were included in the review. 

(2)Exclusion criteria

Studies looking at fetal populations and incidence of antenatal cranial ultrasound findings were excluded, as the purpose of our study was to ascertain if postnatal imaging should be performed. Studies looking at neonates with specific disorders or conditions were excluded (e.g., monochorionic diamniotic twins post laser, infants with congenital heart disease, other congenital anomalies, or metabolic disorders, etc.), as the overall incidence of abnormalities would be higher and not applicable to the general neonatal population. Review articles and case series were excluded, with reference lists of relevant articles crosschecked. 

### 2.2. Population

Moderate-late preterm and term infants, i.e., infants born from 32 weeks’ gestation to term age.

### 2.3. Exposure

FGR/SGA.

### 2.4. Comparator/Control

If available, to appropriately grown infants of the same gestation group.

### 2.5. Outcome Measures

### 2.6. Primary Outcomes

Incidence of any cranial ultrasound abnormalities, assessed before discharge as defined by the authors, in MLPT and term infants affected by FGR or SGA, if possible characterised by abnormality: white matter injury, intraventricular haemorrhage, cerebellar haemorrhage, cerebral haemorrhage, structural abnormalities, etc.

### 2.7. Secondary Outcomes

Review of brain structure development and growth on cranial ultrasounds, of MLPT and term growth-restricted infants prior to discharge, by assessing:
2D- measurements of specific brain structures
cerebellar vermis sizetransverse cerebellar diameter
Review of cerebral artery Dopplers parameters including MCA peak systolic velocity, end diastolic velocity and resistive and pulsatility indices, prior to discharge on cranial ultrasounds of MLPT and term growth-restricted infants

### 2.8. Search Methodology

A systematic search was performed after consultation with a clinical librarian, on three electronic databases PubMed (1996–December 2021), EMBASE (1974–December 2021) and MEDLINE via Ovid (1946–December 2021). Grey literature found through EMBASE was reviewed and included if subgroup analysis was available. Reference lists from relevant review articles and included studies were also manually reviewed to identify potentially relevant studies.

Search terms utilised were: FETAL GROWTH RESTRICTION OR FETAL GROWTH RETARDATION OR INTRAUTERINE GROWTH RESTRICTION OR INTRAUTERINE GROWTH RETARDATION OR SMALL FOR GESTATIONAL AGE OR GROWTH RESTRICTION, AND CRANIAL ULTRASOUND OR BRAIN ULTRASOUND OR ECHOENCEPHALOGRAPHY OR INTRAVENTRICULAR HAEMORRHAGE OR PERIVENTRICULAR LEUKOMALACIA OR INTRACEREBRAL HAEMORRHAGE OR CEREBELLAR HAEMORRHAGE. Terms were mapped and MeSH terms used. Alternate spelling was included. Search methods are detailed in Appendix A.

All studies reporting cranial ultrasonography findings amongst growth-restricted/SGA infants until December 2021 were included.

### 2.9. Study Selection

All references were imported into Covidence Systematic Review Software (Veritas Health Innovation, Melbourne, Australia) for further review and analysis. Duplicates were automatically excluded by the software. Title and abstract screening and cross-checking of the reference lists of relevant studies was performed by two authors (CR, AM) independently. Full text papers were screened by two authors independently (CR, AM) for subgroup analysis of gestation and weight.

### 2.10. Data Extraction 

Data was collected through Covidence with focus on gestation, study type, study numbers, definition of growth restriction, rates of growth-restricted infants, rates of CUAs as total number and percentage for growth-restricted infants and controls if available, types of CUAs identified, key study outcomes and relevant findings to infants either SGA or growth-restricted. Data extraction form is detailed in Appendix A.

### 2.11. Analysis

The papers identified through consensus were included in qualitative and quantitative analysis. Result findings were reported using the PRISMA statement for systematic reviews. 

A random-effects model meta-analyses was performed using Review Manager 5.4.1 (Cochrane Collaboration, Nordic Cochrane Centre, Copenhagen, Denmark) to yield pooled odds ratio (ORs) and 95% Confidence Incidence (CI) for any CUAs and IVH specifically. A *p*-value of <0.05 was considered as significant. Studies comparing MLPT and late preterm to term were pooled as subgroups to test for subgroup differences and identify the source for heterogeneity. Statistical heterogeneity was reported using *I*^2^ values (derived from the *χ*^2^ *Q*-statistic). Significant statistical heterogeneity was an *I*^2^ value greater than 50% or *p* for *χ*^2^ < 0.10.

The Newcastle-Ottawa scale [30] was utilised to assess the quality of studies and the risk of bias by looking at domains of selection, comparability, and outcomes. Refer to Appendix A. For descriptive studies the Joanna Briggs Institute (JBI) Critical Appraisal Checklist for studies reporting prevalence data [31] was utilised. 

Publication bias assessment was not possible as less than 10 studies were included in the meta-analysis. Assessment of the certainty of evidence was performed using the Cochrane Grading of Recommendations, Assessment, Development, and Evaluation (GRADE) methodology.

## 3. Results

### 3.1. Search Results

A literature search was conducted using search terms as described above. Figure 1 depicts the Preferred Reporting Items for Systematic Review and Meta-Analyses (PRISMA) flow diagram for the systematic review. 

The search returned 2027 citations, 329 from MEDLINE, 877 from EMBASE, 763 from PubMed and a further 58 through manual reference checking of review articles. 568 duplicates were removed, 565 were removed through Covidence Systematic Review Software (Veritas Health Innovation, Melbourne, Australia) and another 3 were removed manually. Abstracts from 180 studies were assessed for eligibility, which led to the inclusion of 17 studies that contained relevant data for infants of interest, 14 studies looked at CUAs, 3 studies looked at brain growth and Dopplers. Of the 14 studies looking at CUAs, a further 7 studies looked at FGR verses AGA infants and lent themselves to meta-analysis.

### 3.2. Methodological Quality

The Newcastle-Ottawa scale was used to assess the risk of bias in case–control and cohort studies. [30] Nine studies had a moderate risk of bias [32,33,34,35,36,37,38,39,40], four studies had high risk of bias [41,42,43,44], and one study had a low risk of bias [45]. (Refer to Table 1 and Table 2 for Newcastle-Ottawa scale scoring). The studies that had issues with high risk of bias had issues in all three domains: selection, comparability, and outcomes. Issues around selection criteria, with many studies not mentioning the number of eligible subjects, may have led to response bias. The main issue with comparability was not controlling for weight or other confounding factors. Issues with outcomes identified were that the studies had variable timing of when cranial ultrasounds were preformed, with some only reporting very early ultrasounds leading to possible reporting bias and missing abnormalities that develop or persist. Publication bias assessment was not possible. 

The certainty of the evidence as assessed per GRADE was deemed very low due to the risk of bias, heterogeneity, and observational data.

### 3.3. Study Characteristics

The characteristics of the studies included were six prospective cohort studies and seven retrospective cohort studies and one case–control study. The main findings of articles reviewing MLPT infants are listed in Table 3, Table 4 and Table 5 and those reviewing late preterm and term infants are listed in Table 6, Table 7 and Table 8.

Of the seven studies included in the meta-analysis, three were prospective cohort studies, three were retrospective cohort studies and one was a case–control study.

### 3.4. Cranial Ultrasound Abnormalities in Moderate to Late Preterm Infants

Ten studies assessed CUAs in MLPT growth-restricted infants (see Table 3 and Table 4) [32,34,38,40]. Six studies compared growth-restricted to well grown infants [32,34,37,38,40,45], with four of these studies showing an increased risk of abnormalities in growth-restricted infants [32,34,38,40]. Of these studies seven studies reported only IVH changes on cranial ultrasound with one of these studies only reporting Grade III/IV IVH [46] (see Table 5 for further details).

### 3.5. Cranial Ultrasound Abnormalities in Late Preterm and Term Growth-Restricted Infants

Eight studies assessed CUAs in late preterm and term growth-restricted infants [32,33,34,35,36,39,43,44]. (See Table 6 and Table 7) Three studies compared growth-restricted to well grown infants [32,33,34], with two of these studies showing an increased risk of abnormalities in growth-restricted infants [33,34]. Of these studies, five [32,34,35,39,43] reported IVH alone. (See Table 8 for further details). 

### 3.6. Meta-Analysis

A meta-analysis of seven studies involving 168,136 participants showed an increased risk of any CUAs in FGR infants compared to AGA infants (RR 1.96, [95% CI 1.26–3.04], *I*^2^ = 68%). The certainty of evidence was very low due to non-randomised studies, methodological limitations, and between-studies heterogeneity (Figure 2).

To further test for subgroup differences, a meta-analysis based on gestational age was performed looking at MLPT (32–37^+6^ weeks) and late preterm to term (36–42 weeks) infants, respectively. An increased risk of CUAs was found in the late preterm to term subgroup (RR 4.34, [95% CI 1.18–15.95]) but not in the moderate to late preterm subgroup (RR 1.96, [95% CI 0.96–4.13]) and the test for subgroup differences were not significantly different (*p* = 0.31) (Figure 3). 

When looking at which individual CUAs were reported, four studies reported IVH as the only CUA [32,34,38,40]. The meta-analysis of these 4 studies included 167,060 participants and showed an increased risk of IVH in FGR infants compared to AGA infants (RR 2.40, [95% CI 2.03–2.84], *I*^2^ = 0%). (Refer to Figure 4) There was no heterogeneity. The evidence certainty was low due to non-randomised studies and their methodological limitations. 

No separate data was available to perform a meta-analysis of other CUAs, such as PVL, cerebral or cerebellar haemorrhage.

### 3.7. Brain Structure Growth in SGA Infants

There has been some interest in cerebellar size in infants with growth restriction as a potential prognostic indicator. Huang looked at whether there was a difference in the growth of the cerebellar vermis between AGA and SGA infants [47]. There was a significant difference in the central vermian area and the superior-inferior distance of the cerebellar vermis in infants born 38–41 weeks. Makhoul looked at transverse cerebellar diameter in preterm and term infants that were AGA verses SGA [48]. They found that in both asymmetric and symmetric SGA infants there was not a statistically different size in transverse cerebellar diameter in SGA verses AGA infants. 

### 3.8. MCA Dopplers in Infants with Fetal Growth Restriction

Krishnamurthy et al. looked at MCA Doppler characteristics in 40 infants comparing 20 with FGR and 20 AGA infants [49]. They performed a subgroup analysis and infants > 32 weeks gestation and found a higher MCA peak systolic velocity in FGR infants on day 1 (*p* = 0.01) and 3 (*p* = 0.009) of life. The end diastolic velocity value was only significant on day 3 (*p* = 0.007) of life in FGR infants. In infants > 32 weeks the resistive index and pulsatility index were significantly lower in FGR compared to AGA infants.

## 4. Discussion 

This systematic review and meta-analysis provides an overview of the literature on CUAs in MLPT and term growth-restricted/SGA infants, and to our knowledge is the first of its kind. When reviewing the evidence, our aim was to ascertain what is known about the incidence of CUAs in MLPT and term FGR/SGA/IUGR infants and if routine screening should be considered. It is apparent, that there are still many gaps in the literature. Four main findings of the meta-analysis were that when the data was pooled, infants MLPT and term (>32 weeks) were found to have an increased rate of CUAs in FGR infants when compared with AGA infants. When further looking at subgroup analysis, moderate-late preterm infants born FGR did not have a significant increased rate of CUAs compared with their AGA counterparts; however, late preterm, and term growth-restricted infants had an increased rate of CUAs. When looking at IVH alone, there was an increased rate in infants born FGR versus AGA. Overall, however, the studies presented were heterogenous, of varying quality, and the certainty of evidence was overall low. 

Overall, there was a paucity of studies focusing on CUAs in MLPT and term infants, especially looking at FGR as a risk factor. Many of the studies selected were not set up to answer the clinical question and were often subgroup analyses with small cohorts potentially misrepresenting the true incidence of CUAs in this population group. The definition of FGR varied widely from ICD coding, varying birth weight cut offs (e.g., <3rd, <5th and <10th centiles) and weight and Doppler abnormalities. Even more difficult to interpret is the wide variety in incidence of cranial ultrasound abnormality results, at times greater than those of infants in the extreme and very premature category. While ultrasound is known to be operator-dependent, further challenges around consensus between radiologists and neonatologists when reporting and defining ultrasound findings all add additional complexities when interpreting the data. 

### 4.1. Incidence of Cranial Ultrasound Abnormalities

The incidence of CUAs varied significantly when reviewing the literature. Marsoosi et al. reported the highest rate (33%) of IVH; however, they looked at 6 infants born between 32 to 33^+6^ weeks and found that 2 of these infants had IVH. Such a small sample size brings into question the validity of such a high incidence of abnormalities. Ballardini reported the second highest incidence of CUAs with 19.4% in <3rd centile infants born between 33^+0^–36^+6^ weeks; PVL was not defined, and increased incidence could be due to potential over-reporting of white matter injury [45,50]. Gilbert reported the lowest incidence of IVH in FGR MLPT infants, with an incidence of 0.4% in infants born at 36 weeks [34]. While this was a comprehensive and extensive database review, the difficulty with interpreting these findings is that FGR was defined by ICD coding. Baschat reported the second lowest incidence in this cohort of 1.3% in infants born 32–32^+6^ weeks, however, they only reported grade III-IV IVH [41]. Understandably these are the most significant changes on ultrasound linked with long-term neurodevelopmental complications; however, more recent evidence is suggesting that even grade I and II IVH may indeed have long-term complications due to microstructural changes in periventricular and subcortical white matter [51].

Similarly in the studies looking at late preterm and term infants the incidence of abnormalities varied widely. Mercuri studied a very small cohort of growth-restricted infants that found 7 of the 10 infants with CUAs, the specific findings of which were not listed and is therefore difficult to interpret. Surprisingly the most common abnormality documented was ischaemic lesions with 8% infants having periventricular and thalamic densities. Starcevic’s rate of CUAs of 53.37% is also hard to explain. Infants were screened on day 1, 3 and 7 and while PVL changes may have been overestimated, the rate of IVH was also significantly higher than recorded rates of very preterm infants and whether differences around perinatal care may affect these rates so drastically is questionable [51].

Our findings of such variability in incidence reported in the literature is not unique, a recent systematic review by Boswinkel looked at brain injury and altered brain development in moderate-late preterm infants, by reviewing a mix of cranial ultrasound and MRI results and found a wide incidence of CUAs (0.0 to 23.5%) [50]. There was noted to be paucity of high-quality studies set up to answer the incidence of brain injury in this population and heterogenous study cohorts.

### 4.2. FGR versus AGA Cranial Ultrasound Result Comparison

When comparing FGR infants with AGA infants MLPT infants did not have a significant risk of CUAs, whereas late preterm and term FGR infants may have an increased risk of CUAs. (See Figure 3). When looking at MLPT infants, four of six studies identified showed an increased risk of CUAs in MLPT, three [32,38,40] used SGA as a surrogate for FGR and one used ICD coding [34]. When looking at late preterm and term infants, two of the three studies identified showed an increased rate of CUAs. One study used ICD coding, a further used a cut-off of weight < 10th centile and yet another referred to medical notes.

These results may suggest that infants born MLPT may indeed have higher rates of CUAs than once thought, additionally using SGA as a surrogate marker may not accurately capture those infants affected by FGR and render any difference between cohorts as insignificant. Of the two studies that found FGR infants were not at increased risk of CUAs, Ballardini and colleagues overall found a high rate of CUAs in their cohort and concluded that screening should be considered for the MLPT population, especially the 33–34-week cohort [45]. While Medina-Alva et al.’s purpose was not to look at the incidence of CUA in our intended population, and indeed the rate of major abnormalities on CUA was slightly greater in those infants born AGA compared to SGA [37]. This discrepancy may be explained by the perinatal complications surrounding infants born MLPT where there may be chorioamnionitis or incomplete steroid use, compared with the planned and controlled delivery of SGA infants, alternatively it may be due to the use of SGA as a cut off and not truly identifying FGR infants.

When looking at late preterm and term growth-restricted infants, the two studies that showed an increased rate of CUA in growth-restricted infants were significant outliers with their reported incidence of abnormalities [33]. Conversely, Berger found that SGA infants at term were not at increased risk of CUA, this may be explained by a heterogenous cohort of healthy small and growth-restricted infants [33].

While the focus has traditionally been on IVH and white matter injury; our understanding of the CUA seen in MLPT and term infants, especially those that are growth-restricted, is limited. Our focus may need to shift from our typical patterns of brain injury seen in prematurity, to looking at brain growth of specific structures, Doppler parameters, or consideration of MRI to further delineate more subtle patterns of brain injury. 

Overall, MLPT and term growth-restricted infants are potentially no longer the low-risk cohort we once believed. Various other risk factors have been noted to contribute towards intracranial pathology in the late preterm population—including intubation, low cerebroumbilical ratios, hypothermia, low APGAR scores, head circumference below the 3rd percentile at birth, low venous/arterial pO2 and pH [32,36,45,52]. Given the association between increased rates of intracranial pathology and low arterial/venous pH and pO2, and use of ventilatory supports, there may be suggestion that hypoxic insult in MLPT infants plays an important indicator contributing towards abnormal cranial ultrasonography results. Starcevic et al. also described low cerebroumblical ratios (CUR), umbilical resistance index and low birthweight to be related to adverse functional outcomes indicating clinical significance. CURs ≤ 1 are an important marker of fetal compromise and have been linked to adverse perinatal outcomes with strongest association in low birthweight babies [53]. Khazardoost also found that absent or reversed end-diastolic velocity in the UA and DV was associated with an increased rate of CUAs [54]. Although it is not possible to fully ascertain the benefit of cranial ultrasonography screening upon all growth-restricted infants, risk stratification may be a useful tool in the development of screening guidelines. The presence of small-for-gestational age, low cerebroumbilical ratios and indicators of birth asphyxia may act as important tools to identify high risk infants that may benefit from cranial ultrasound screening.

## 5. Limitations

To date, there have been limited studies focusing on MLPT and term growth-restricted infants and minimal data evaluating long-term developmental outcomes associated with abnormal cranial ultrasonography results. Inconsistent definitions of growth restriction and the use of surrogate markers to identify growth restriction present a challenge in recognition of the pathologically growth-restricted fetus and may attribute to the discrepancies seen in the literature. Furthermore, this review carries multiple limitations including significant heterogeneity of studies, methodologically weak studies not powered to answer our research question and low certainty of the evidence. When addressing heterogeneity, multiple aspects may have contributed to this including differences in risks and screening, pooling of CUAs data together and gestation subgroups. There was no heterogeneity when looking at IVH alone, suggesting that pooling all CUAs abnormalities may be one possible source of heterogeneity. There were also considerable differences in gestational age among studies and despite no difference between the subgroups, merely testing two subgroups may not be sufficient. Additionally, studies were observational in nature, retrospective and with differing study aims, with data often being extrapolated to examine specific subgroups of interest. Studies were of varying designs, with different cohort years and protocols for and timing of cranial sonography. Ultrasound screening protocols varied between studies, and the timing and frequency may play an important role in the detection of intracranial abnormalities. Eight studies alone focused on IVH and no other CUAs, which may not in fact be the focus when imaging MLPT and term infants, conversely pooling all CUAs lead to significant heterogeneity. In addition, cohorts were grouped in moderate-late preterm and late preterm-term infants as many studies did not gave further subgroup data available or had overall small cohorts. This further complicates the interpretation of data infants as each gestational group is a unique entity with often different reasons for delivery and different risks of brain injury. 

### Implications for Future Research 

Dedicated cohort studies with multivariable analysis are needed to fully evaluate the benefits and prognostic value of cranial ultrasonography to ascertain the need for and timing of a cranial ultrasound screening protocol in this infant population. Further research on brain growth and MCA Dopplers in further characterising structural changes associated with FGR that may be useful for prognostication. Long-term follow-up studies to ascertain the significance of CUAs in the MLPT and term growth-restricted cohort are warranted. The development and evaluation of risk stratification tools may play an important role in developing neuroimaging guidelines for identification of infants that are at greater risk of neurodevelopmental abnormalities.

## 6. Conclusions

The incidence of CUAs in MLPT and term growth-restricted infants varied widely between studies. The findings from the meta-analysis suggest the risk of CUAs and IVH may indeed be increased in FGR/SGA infants when compared with infants not affected by FGR, however the evidence is of low to very low certainty. It is increasingly apparent that these cohorts have long-term neurological sequelae based on long term follow up studies; however, the incidence, pattern of brain injury and potential abnormalities on neuroimaging in addition to their long-term outcomes have been poorly characterised thus far.

## Figures and Tables

**Figure 1 brainsci-12-01713-f001:**
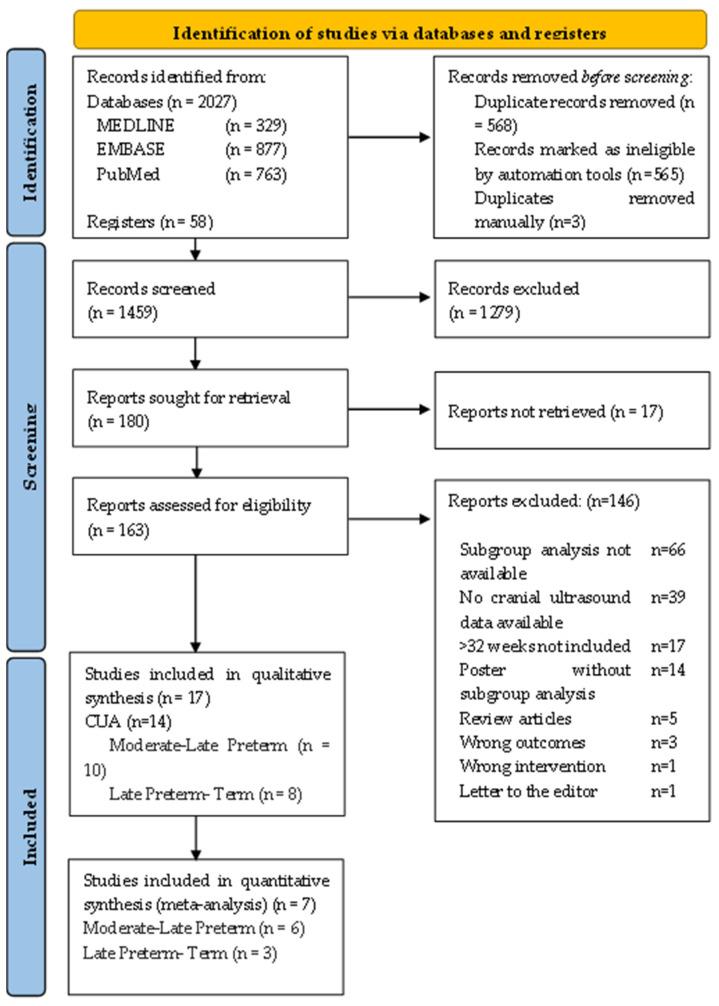
PRISMA flow diagram.

**Figure 2 brainsci-12-01713-f002:**
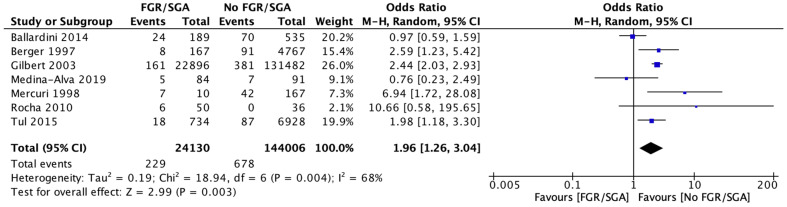
Forest plot comparing the risk of any cranial ultrasound abnormality among babies with fetal growth restriction compared to babies without fetal growth restriction among all infants more than 32 weeks.

**Figure 3 brainsci-12-01713-f003:**
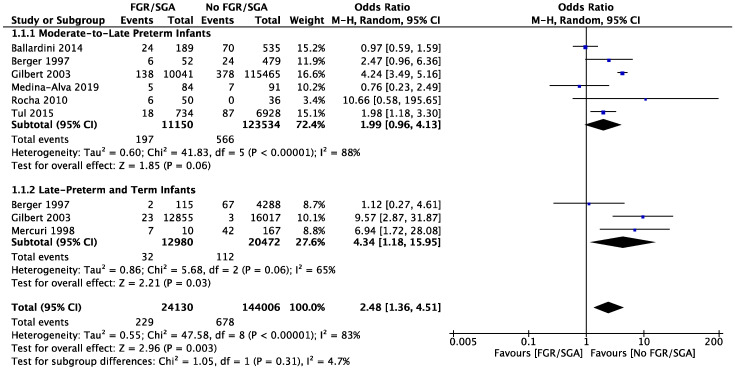
Forest plot comparing the risk of any cranial ultrasound abnormality among moderate to late preterm and late preterm to term subgroups with or without fetal growth restriction.

**Figure 4 brainsci-12-01713-f004:**
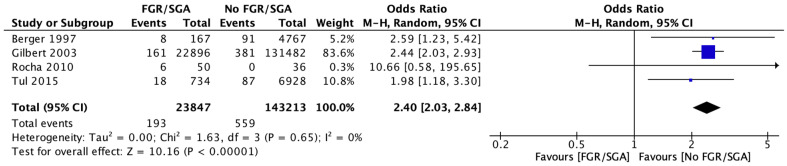
Forest plot comparing the risk intraventricular haemorrhage among babies with fetal growth restriction compared to babies without fetal growth restriction among all infants more than 32 weeks.

**Table 1 brainsci-12-01713-t001:** Newcastle Ottawa Scale: COHORT STUDIES.

	Selection				Comparability	Outcome			Total
	Representativeness of the Exposed Cohort	Selection of the Non-Exposed Cohort	Ascertainment of Exposure	Demonstration That Outcome of Interest Was Not Present at Start of Study	Comparability of Cohorts on the Basis of the Design or Analysis	Assessment of Outcome	Was Follow-Up Long Enough for Outcomes to Occur	Adequacy of Follow Up of Cohorts	
Berger, 1997 [32]	★	★	★	-	★	★	-	★	★★★★★★ Moderate Risk
Mercuri, 1998 [33]	-	★	★	-	★	★	-	-	★★★★ Moderate Risk
Gilbert, 2003 [34]	★	★	★	-	★	★	-	★	★★★★★★ Moderate Risk
Baschat, 2007 [46]	-	-	★	★	-	★	-	-	★★★ High Risk
Valcamonico, 2007 [42]	-	-	★	★	-	★	-	-	★★★ High Risk
Marsoosi, 2012 [35]	-	-	★	★	★	★	★	★	★★★★★★ Moderate Risk
Ballardini, 2014 [45]	★	★	★	★	★	★	-	★	★★★★★★★ Low Risk
Tul, 2015 [40]	★	★	★	-	★	★	-	★	★★★★★★ Moderate Risk
Starcevic, 2016 [36]	-	-	★	★	-	★	★	★	★★★★★ Moderate Risk
Krishnamurthy, 2017 [44]	-	-	★	-	-	★	-	★	★★★ High Risk
Stimac, 2019 [43]	-	-	★	-	-	★	-	-	★★ High Risk
Medina-Alva, 2019 [37]	-	-	★	★	-	★	★	-	★★★★ Moderate Risk
Turcan, 2020 [39]	★	★	★	-	★	★	★	-	★★★★★★ Moderate Risk

One star was awarded for each domain, except for comparability, where a maximum of two stars could be awarded. Total scores ranged from 0–9. Low risk of bias studies were awarded 7–9 stars, moderate risk of bias studies were awarded 4–6 stars and high risk of bias studies were awarded 0–3 stars.

**Table 2 brainsci-12-01713-t002:** Newcastle Ottawa Scale: CASE CONTROL STUDIES.

	Selection				Comparability	Outcome			Total
	Is the Case Definition Adequate	Representativeness of the Cases	Selection of Controls	Definition of Controls	Comparability of Cases and Controls on the Basis of the Design or Analysis	Ascertainment of Exposure	Same Method of Ascertainment for Cases and Controls	Non-Response Rate	
Rocha, 2010 [38]	★	-	-	★	★	★	★	-	★★★★★ Moderate Risk

One star was awarded for each domain, except for comparability, where a maximum of two stars could be awarded. Total scores ranged from 0–9. Low risk of bias studies were awarded 7–9 stars, moderate risk of bias studies were awarded 4–6 stars and high risk of bias studies were awarded 0–3 stars.

**Table 3 brainsci-12-01713-t003:** Studies including cranial ultrasonography abnormalities in moderate-late preterm growth-restricted and SGA infants.

Author, Year	Gestation	Study Total	Study Design	Aims	Key Findings of Study	Relevant Findings Related to FGR or SGA
Berger, 1997 [32]	24^+0^–43^+0^ weeks	5286	Prospective cohort study	To examine the incidence of brain injury on cranial ultrasound compared with obstetric risk factors	The most frequent abnormality was IVH, with the incidence increasing with decreasing gestational age	Growth restriction and acidosis (pH ≤ 7.29) on arterial cord gases was associated with an increased risk of IVH That rate of IVH in FGR infants born 35–37 weeks was 11.5%
Gilbert, 2003 [34]	26^+0^–41^+0^ weeks	1,347,788	Retrospective cohort study	To examine incidence of FGR and associated neonatal outcomes	Until 28 weeks, prematurity associated with adverse neonatal outcomes (RDS, IVH, NEC, CHA) was largely unaffected by FGR	Of the FGR Infants born between 34–39 weeks there was a statistically significant increase in the rate of IVH compared to AGA counterparts
Valcamonico, 2007 [42]	24^+0^–34^+6^ weeks	183	Prospective cohort study	To evaluate morbidity and long-term neurological outcomes in extremely low birthweight infants (i.e., <1000 g)	Extremely low birth weight increases the risk of perinatal morbidity and neonatal morbidity The most significant factor for long-term neurological outcomes was gestational age	Subgroup analysis of 10 infants born between 32–34 weeks gestation had 10% (n = 1) infant with IVH and 0 with PVL
Baschat, 2007 [46]	24^+0^–32^+6^ weeks	604	Prospective cohort study	To determine morbidity and mortality in growth-restricted infants with early onset placental dysfunction	Gestational age was the most significant predictor (*p* < 0.005) of survival until 26^+6^ weeks and intact survival until 29^+2^ weeks. Beyond 29^+2^ weeks and in infants > 600 g, ductus venosus Doppler and cord artery pH was predictive of neonatal mortality (*p* < 0.001) and Doppler alone was predictive of intact survival.	Of 76 infants born growth-restricted at 32 weeks 1.3% (n = 1) had a grade III-IV IVH
Rocha, 2010 [38]	34^+0^–36^+6^ weeks	86	Case-control study	To compare neonatal outcomes between FGR and AGA infants	Late-preterm FGR infants require longer hospitalisation and are at higher risk of IVH and hypoglycaemia compared to AGA infants	Greater rates of IVH * seen in FGR population vs. AGA infants (12% vs. 0%, *p = 0.037*)
Marsoosi, 2012 [35]	23^+0^–40^+6^ weeks	41	Prospective cohort study	To determine if there is a correlation between Doppler indices and IVH and perinatal mortality in pregnancies affected by FGR	Infants with AREDF had a 5 times greater chance of developing IVH The risk of IVH was associated with gestational age at delivery, birth weight, and acidosis	The rate of IVH was 33% in infants born between 32–33^+6^ weeks and 12.5% in infants born between 34–35^+6^ weeks
Ballardini, 2014 [45]	33^+0^–36^+6^ weeks	724	Retrospective cohort study	To determine the number of neonates with abnormal cranial ultrasounds and evaluate universal ultrasound screening	Infants born 33^+0^–34^+6^ weeks were four times more likely to have CUAs compared with those born at 35^+0^–36^+6^ weeks A postnatal head circumference < 3rd centile, need for ventilation or surfactant, low APGAR at the 5th minute of life and neurological abnormalities were associated with an increase rate of CUAs	No significant increase in rate CUAs in SGA infants with birth weights < 10th or <3rd centiles.
Tul, 2015 [40]	24^+0^–36^+6^	159,774	Retrospective cohort study	To compare short term outcomes and morbidity of SGA and AGA infants by gestational age and maternal, antenatal and birth details	Infants born > 30 weeks and SGA had worse 5 min APGAR scores, and higher rates of IVH, RDS, neonatal death and ventilation compared with infants that were AGA.	Infants born 33–36^+6^ weeks and SGA had a 2-times increase rate of IVH compared to AGA infants. (2% (n = 18) vs. 1.3% (n = 87) OR 2 (1.2–3.3 95%CI)
Stimac, 2019 [43]	23^+0^–>37 weeks	2676	Retrospective cohort study	To examine the effect on neonatal outcomes of gender in SGA infants	Both female and male infants had similar rates of RDS, IVH and admission to intensive care	The rate of IVH documented in SGA infants born 33–36^+6^ weeks was 5.1% (8 of 158 infants)
Medina-Alva, 2019 [37]	24–36 weeks	414	Prospective cohort study	To assess the combined prognostic value of neurological examination, head circumference and cranial ultrasounds for neurodevelopmental delay in very low birth weight (VLBW, <1500 g) preterm infants	A combination of microcephaly and major ultrasound abnormalities (brain infarction, parenchymal haemorrhage, grade 3 or 4 IVH, post-haemorrhagic hydrocephalus with ventricular index > 14mm and PVL/or periventricular cysts) showed the highest positive predictive value (100%; 95% CI, 51%–100%) for poorer neurological outcomes.	5.9% of SGA infants born between 32–36 weeks had major ultrasound abnormalities, vs. 7.7% of infants that were well grown

Grade I IVH was the only abnormality documented. AREDF = absent/reversed umbilical diastolic flow, AGA = appropriately grown infants, CHA = hospital charges, C/U = cerebroumbilical, FGR = Fetal growth restriction/Intrauterine Growth Restriction, IVH = Intraventricular haemorrhage, LOS = length of stay, NEC = Necrotising enterocolitis, OR = odds ratio, PVL = Periventricular Leukomalacia, RDS= Respiratory Distress Syndrome, SGA = Small for gestational age, US = Ultrasound, VLBW = Very low birth weight.

**Table 4 brainsci-12-01713-t004:** Study Characteristics and Incidence of CUAs Reporting on Moderate-Late Preterm Infants.

First Author	Subgroup Gestation	Subgroup Total	Definition of FGR/SGA Infants	Number of FGR/SGA Infants	Study Population Incidence of CUAs	CUAs Incidence in Non FGR/SGA Infants	CUAs Incidence in FGR/SGA
Berger [32]	35^+0^–37^+6^	531	<10th Centile	52	3.6%	5%	11.5%
Gilbert [34]	32^+0^–32^+6^ 33^+0^–33^+6^ 34^+0^–34^+6^ 35^+0^–35^+6^ 36^+0^–36^+6^	5891 9994 17,843 30,247 51,490	Based on ICD coding at discharge	1740 1874 2015 2157 2255	Not reported	1.8% 0.9% 0.4% 0.2% 0.1%	2.7% 1.5% 1.3% 1.3% 0.4%
Valcamonico [42]	32^+0^–34^+6^	10	As infants < 1000 g—based on preterm growth charts would be <3rd centile	10	28.9%	-	10%
Baschat [46]	32^+0^–32^+6^	76	AC < 5th centile + elevated UAPI	76	15.2%	-	1.3%
Rocha [38]	34^+0^–36^+6^	86	<10th centile	50	7%	0%	12%
Marsoosi [35]	32^+0^–33^+6^ 34^+0^–35^+6^	6 8	EFW + AC < 10th percentile + UAPI + UARI > 2 SD	6 8	17.1%	-	33% 12.5%
Ballardini [45]	33^+0^–36^+6^	724	<10th Centile <3rd Centile	189 31	13%	-	12.6% 19.4%
Tul [40]	24^+0^–36^+6^	7662	<10th centile based on local birth weight data	734	3.9%	1.37%	2.5%
Stimac [43]	33^+0^36^+6^	158	<5th centile based on local birth weight data	158	3.8%	-	5.1%
Medina-Alva [37]	32^+0^–36^+6^	175	<10th centile	84		7.7%	5.9%

AC = Abdominal circumference, CUAs = Cranial ultrasound abnormalities, EFW = Estimated fetal weight, ICD = International Classification of Diseases, UAPI = Umbilical artery pulsatility index, UARI = Umbilical artery resistance index, SD = Standard deviation. Results in bold have a statistically significant *p* value.

**Table 5 brainsci-12-01713-t005:** Reported Abnormalities in MLPT FGR infants.

Berger, 1997 [32]	IVH
Gilbert, 2003 [34]	IVH
Valcamonico, 2007 [42]	IVH (classification by Volpe) PVL (classification by Pierrat)
Baschat, 2007 [46]	Grade III/IV IVH
Rocha, 2010 [38]	IVH
Marsoosi, 2012 [35]	IVH
Ballardini, 2014 [45]	IVH (classification by Volpe) PVL (classification by De Vries) Agenesis of corpus callosum Other haemorrhages Enlarged cisterna magna Hyperechogenicity of thalami Other major anomalies of the brain
Tul, 2015 [40]	IVH
Medina-Alva, 2019 [37]	Brain infarction Parenchymal haemorrhage Grade III/IV IVH Post-haemorrhagic hydrocephalus with ventricular index > 14mm PVL/or periventricular cysts
Stimac, 2019 [43]	IVH

**Table 6 brainsci-12-01713-t006:** Studies including cranial ultrasonography in late preterm and term SGA/FGR infants.

Author, Year	Gestation	Study Total	Study Design	Aims	Key Findings of Study	Relevant Findings Related to FGR or SGA
Berger, 1997 [32]	24^+0^–43^+0^ weeks	5286	Prospective cohort study	To examine the incidence of brain injury on cranial ultrasound compared with obstetric risk factors	The most frequent abnormality was IVH, with the incidence increasing with decreasing gestational age	1.7% (n = 2) of 115 infants born < 10th Centile at 38–43 weeks were documented to have IVH
Mercuri, 1998 [33]	36^+0^–42^+0^ weeks	177	Prospective cohort study	To evaluate cranial ultrasounds and neurological examinations in ‘normal’ neonates and correlate with perinatal factors	Ultrasound abnormalities were present in 20% (n = 35) of the infants studied The most common finding were ischaemic lesions (periventricular and thalamic densities) seen in 8% (n = 15); IVH in 5% (n = 9) of infants	7 of the 10 infants that were FGR had abnormal ultrasound findings (specific findings not listed)
Gilbert, 2003 [34]	26^+0^–41^+0^ weeks	1,347,788	Retrospective cohort study	To examine incidence of FGR and associated neonatal outcomes	Until 28 weeks, prematurity associated with adverse neonatal outcomes (RDS, IVH, NEC, CHA) was largely unaffected by FGR	Of the FGR Infants born between 34–39 weeks there was an increased rate of IVH compared to their well grown counterparts
Marsoosi, 2012 [35]	23^+0^–40^+6^ weeks	41	Prospective cohort study	To determine if there is a correlation between Doppler indices, IVH and perinatal mortality in pregnancies affected by FGR	Infants with AREDF had a 5 times greater chance of developing IVH The risk of IVH was associated with gestational age at delivery, birth weight, and acidosis	Of the 13 infants born ≥ 36 weeks, no infants developed IVH
Starcevic, 2016 [36]	≥37 weeks	60	Retrospective cohort study	To ascertain if infants born at term, diagnosed with late-onset FGR, have CUAs and abnormal neurological examinations and to assess predictive values of umbilical cord gases and umbilical Doppler indices	Abnormal umbilical Doppler indices were more predictive than cord blood gases of neurological outcomes, with C/U being most predictive	53.37% (n = 32) of infants had had CUAs detected 38.33% (n = 23) had IVH, and 15.0% (n = 9) had PVL
Krishnamurthy, 2017 [44]	35^+0^–43^+0^ weeks	415	Retrospective cohort study	To determine frequency of cranial US screening and incidence of abnormal ultrasonography in SGA infants	No significant increased yield when screening <3rd vs. < 10th centile infants. The majority of infants had positive minor findings (e.g., grade 1–2 IVH) The rest of the infants with abnormal cranial ultrasounds were known antenatal findings or those with significant risk factors (e.g., HIE)	12.8% of infants with a birth weight < 10th centile had CUAs and 11% of infants with a birth weight < 3rd centile had CUAs (no control group)
Stimac, 2019 [43]	23^+0^–>37 weeks	2676	Retrospective cohort study	To examine the effect on neonatal outcomes of gender in SGA infants	Both female and male infants had similar rates of RDS, IVH and admission to intensive care	The rate of IVH documented in SGA infants born > 37 weeks was 0.5%
Turcan, 2020 [39]	23^0^–41^0^ weeks	1405	Retrospective cohort study	To compare the rate of short-term complications of preterm infants born with and without low birth weight and term low birth weight infants	Infants in the preterm SGA cohort had the highest frequency of neonatal complications	Subgroup analysis of 206 infants born with ‘low birth weight’ (< 2SD from curve for gestation) at 37–41 weeks showed 0.4% (n = 1) had cerebral haemorrhage and 1.4% (n = 3) had IVH

AREDF = absent/reversed umbilical diastolic flow, CHA = hospital charges, C/U = cerebroumbilical, FGR = Fetal growth restriction/Intrauterine Growth Restriction, HIE = Hypoxic Ischaemic Encephalopathy, IVH = Intraventricular haemorrhage, NEC = Necrotising enterocolitis, PVL = Periventricular Leukomalacia, RDS = Respiratory Distress Syndrome, SGA = Small for gestational age, US = Ultrasound.

**Table 7 brainsci-12-01713-t007:** Study Characteristics and Incidence of CUAs Reporting on Late Preterm and Term Infants.

First Author	Subgroup Gestation	Subgroup Total	Definition of FGR/SGA Infants	Number of FGR/SGA Infants	Study Population Incidence of CUAs	CUAs Incidence in Non FGR/SGA Infants	FGR/SGA Gestation Incidence of CUAs
Berger [32]	38^+0^–43^+0^	4403	< 10th Centile	115	3.6%	1.6%	1.7%
Mercuri [33]	36^+0^–42^+0^	177	As documented in patient clinical notes	10	19.7%	-	70%
Gilbert [34]	37^+0^–37^+6^ 38^+0^–38^+6^ 39^+0^–39^+6^ 40^+0^–40^+6^ 41^+0^–41^+6^	106,939 220,170 351,279 340,887 202,058	Based on ICD coding at discharge	2386 2498 2598 2679 2694	Not reported	0.1% 0% 0% 0% 0%	0.5% 0.2% 0.1% 0.1% 0.1%
Marsoosi [35]	36^+0^–40^+6^	12	EFW + AC < 10th percentile + UAPI + UARI > 2 SD	12	17.1%	-	0%
Starcevic [36]	≥37^+0^	-	EFW < 10th centile + elevated UARI	60	53.4%	-	53.4%
Krishnamurthy [44]	35^+0^–43^+0^	415	<10th centile <3rd centile	201 214	12.8% 11%	-	12.8% 11%
Stimac [43]	>37	2363	<5th centile based on local birth weight data	2363	3.8%	-	0.5%
Turcan [39]	37^0^–41^0^	206	<2 SD for weight	206	10%	-	1%

AC = Abdominal circumference, CUAs = Cranial ultrasound abnormalities, EFW = Estimated fetal weight, ICD = International Classification of Diseases, UAPI = Umbilical artery pulsatility index, UARI = Umbilical artery resistance index, SD = Standard deviation.

**Table 8 brainsci-12-01713-t008:** Reported Abnormalities in LPT/Term FGR infants.

Berger, 1997 [32]	IVH
Mercuri, 1998 [33]	Periventricular densities Unilateral thalamic densities Focal asymmetrical ventricular dilatation, paramedian or choroid cysts. IVH (classification by de Vries) White matter haemorrhagic changes Periventricular densities Full choroid
Gilbert, 2003 [34]	IVH
Marsoosi, 2012 [35]	IVH
Starcevic, 2016 [36]	PVL (classification by Pidcock) IVH (classification by Volpe)
Krishnamurthy, 2017 [44]	Cysts: in the caudothalamic groove, sub-ependymal, interhemispheric, or posterior fossa IVH (Grade I/II) Echogenicity in periventricular area (PVE) or basal ganglia Agenesis or dysgenesis of corpus callosum Mild ventricular dilatation Hydrocephalus Arnold Chiari malformation Cerebellar hypoplasia Colpocephaly
Turcan, 2020 [39]	IVH
Stimac, 2019 [43]	IVH

## Data Availability

Detailed search data available from authors on request.

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
