# Peer review of "Cranial Ultrasound Abnormalities in Small for Gestational Age or Growth-Restricted Infants Born over 32 Weeks Gestation: A Systematic Review and Meta-Analysis"

_brainsci, 2022, doi:10.3390/brainsci12121713_

Round 1
Reviewer 1 Report
Comprehensive review, all required literature references.
Attempt at answering a very important clinical question.
In results section will highlight the finding of higher clinical significance of CUS in near term SGA/FGR newborns, compared to late preterm newborns.
Author Response
Comprehensive review, all required literature references.
Attempt at answering a very important clinical question.
In results section will highlight the finding of higher clinical significance of CUS in near term SGA/FGR newborns, compared to late preterm newborns.
Thank you.
We have highlighted the finding of higher CUAs in near term infants compared to late preterms in the results section.
Reviewer 2 Report
I appreciate the authors presenting this review article. As the author mentioned that multiple limitations including significant heterogeneity of studies, methodologically weak studies not powered to answer our research question and low certainty of the evidence.........There is so much uncertainty that the accuracy of the results is questionable.
Author Response
I appreciate the authors presenting this review article. As the author mentioned that multiple limitations including significant heterogeneity of studies, methodologically weak studies not powered to answer our research question and low certainty of the evidence.........There is so much uncertainty that the accuracy of the results is questionable.
Thank you.
This is highlighted in the discussion.
Reviewer 3 Report
The authors have created a research site in a medical field of interest Cranial ultrasound abnormalities in small for gestational
ages. The synthesis is well structured and of interest to the medical scientific community.
The authors have included in the synthesis a lot of research in the field, and the information is well synthesized and structured. Also, the authors propose some solutions for the continuation of development researches in the addressed field.
The work is interesting and deserves to be published.
Author Response
The authors have created a research site in a medical field of interest Cranial ultrasound abnormalities in small for gestational
ages. The synthesis is well structured and of interest to the medical scientific community.
The authors have included in the synthesis a lot of research in the field, and the information is well synthesized and structured. Also, the authors propose some solutions for the continuation of development researches in the addressed field.
The work is interesting and deserves to be published.
Thank you for your kind comments.